# Regulation of atmospheric circulation controlling the tropical Pacific precipitation change in response to $CO_2$ increases

Byung-Ju Sohn[1], Sang-Wook Yeh[2], Ahreum Lee [1] & William K.M. Lau[3]

The spatial pattern of precipitation responses to $CO_2$ concentration increases significantly influences global weather and climate variability by altering the location of tropical heating in a warmer climate. In this study, we analyze the Coupled Model Intercomparison Project Phase 5 (CMIP5) climate model projections of tropical Pacific rainfall response to quadrupled increase of $CO_2$. We found that the precipitation changes to the $CO_2$ concentration increase cannot be interpreted by a weakening or strengthening of large-scale east–west coupling across the tropical Pacific basin, i.e., Walker circulation. By calculating the water vapor transport, we suggest instead that different responses of the Walker and Hadley circulations to the increasing $CO_2$ concentration shape the details of the spatial pattern of precipitation in the tropical Pacific. Therefore, more regionally perturbed circulations over the tropical Pacific, which is influenced by the mean state change in the tropical Pacific and the enhanced precipitation outside the tropical Pacific, lead to greater increases in precipitation in the western equatorial Pacific as compared to the eastern tropical Pacific in a warmer climate.

[1] School of Earth and Environmental Sciences, Seoul National University, Seoul 08826, Korea. [2] Department of Environmental Marine Science, Hanyang University, Ansan 15588, Korea. [3] Earth System Science Interdisciplinary Center, University of Maryland, College Park, MD 20742, USA. Correspondence and requests for materials should be addressed to S.-W.Y. (email: swyeh@hanyang.ac.kr)

The atmospheric concentration of carbon dioxide ($CO_2$), which is well known as a strong greenhouse gas, has been gradually increasing since the mid-19th century and will continue to increase in the near future[1,2]. In addition to a rising of global mean surface temperature, an increase of $CO_2$ concentration leads to changes in a number of atmospheric and oceanic variables in terms of their spatial pattern and intensity, as well as frequency[1,3–9]. Among them, the tropical Pacific precipitation changes due to the increase of $CO_2$ concentration have received much attention because precipitation is one of the most important parameters of the hydrological cycle, as well as the ecological environment[10–15]. In particular, the spatial pattern of precipitation in the tropical Pacific under global warming significantly influences global weather and climate variability by altering the location of tropical heating[8,15–20]. Furthermore, the amount of precipitation in the central-to-eastern tropical Pacific can be used as an index to represent the intensity of El Niño and Southern Oscillation (ENSO)[21,22], which is the most dominant variability of sea surface temperature (SST)[7,23]. In addition, the influence of the eastern tropical Pacific state, including mean SST and precipitation, on global climate is equally important[24,25]. Therefore, it is crucial to understand the details of the physical processes playing key roles in determining the spatial pattern of tropical Pacific precipitation in the warmer climate, due to the increase of greenhouse gas concentration.

Most previous studies examining the precipitation change in the tropical Pacific under the warmer climate are based upon the analysis of the Coupled Model Intercomparison Projection Phase 5 (CMIP5) climate model results. Precipitation change in a warmer climate is often characterized as a wet-get-wetter/dry-get-drier pattern[16,17] or a warmer-get-wetter pattern[18,26]. While the wet-get-wetter/dry-get-drier pattern thermodynamically interprets an increasing atmospheric water vapor content and transport following the Clausius–Clapeyron relation, the warmer-get-wetter mechanism dynamically interprets the cause of precipitation pattern change in the tropics. Specifically, the warmer-get-wetter mechanism indicates that the precipitation changes in the tropics are positively correlated with the spatial deviations of SST warming relative to the tropical mean SST, because the moist instability is determined by the relative SST changes. Although these two mechanisms reasonably well explain precipitation changes in a warmer climate, future projection of rainfall patterns is still a challenging issue[27,28]. The scientific community requires more insight on the mechanisms of precipitation changes to precisely project future climate. Using CMIP5 $CO_2$ quadrupling experiment results, we shed a new insight on interpreting precipitation change in the tropical Pacific based on two main atmospheric circulations, i.e., Walker and Hadley circulations. Previous studies have emphasized the SST regulations determining the precipitation pattern. Furthermore, recent studies are paying more attention to the respective roles of Walker and Hadley circulation, determining the structure of divergence/convergence driving the pattern of precipitation changes over the tropical Pacific in a warmer climate[29–33]. In this study, we focus more on the atmospheric regulation controlling the tropical Pacific precipitation pattern in a warmer climate, in response to $CO_2$.

We mainly use outputs from 21 CMIP5 model simulations based on a 140-year experiment with a prescribed 1% per year increase in $CO_2$ concentration (i.e., the quadrupling experiment of $CO_2$ concentration) (Supplementary Table 1)[34], excluding other greenhouse gases (GHGs) and aerosols. To underpin the impact of $CO_2$ concentration increase on the tropical circulation, the 1% $CO_2$ experiment outputs from the 21 climate models are averaged for the multi-model ensemble mean, representing the climate variability forced by $CO_2$ change. We consider the first 20-year (1–20 years) simulations as a control run. Since an annual 1% increase of $CO_2$ concentration results in approximately a quadrupling of $CO_2$ concentrations in the last 20 years of the 140-year analysis period (i.e., 121–140 years), the difference between the two periods (1–20 years vs. 121–140 years) is considered to be the precipitation response to quadrupling of $CO_2$ concentration. We also analyze outputs from doubling and tripling $CO_2$ concentration experiments from 21 CMIP5 model simulations (see the Methods section) and 21 CMIP5 preindustrial simulations (Supplementary Table 1) to ensure that the results based on the quadrupling experiment of $CO_2$ concentration are representative. Note that the responses due to doubling and tripling of $CO_2$ concentration were similar to those from the quadrupling except with stronger signals in the quadrupling of $CO_2$. Thus, in this study, we mainly provide results from a quadrupling of $CO_2$ concentration (hereafter referred to as $QCO_2$).

## Results

**Mass overturning circulation over the tropics.** In order to understand the precipitation change over the tropics in a warmer climate, we focus on the mass overturning circulation. We first examine the total water vapor transport (hereafter, $\mathbf{Q}$)[35] (see Methods section) because it takes place mostly in the boundary layer, and establish a path mostly from the sinking region in the subtropics to the ascending region over the Equator. Along with $\mathbf{Q}$, we also use the divergent wind field at 200-hPa level to describe the upper branch of the circulations linked to the lower-level circulation branch. However, considering that water vapor flux is also a function of water vapor change itself, the effective boundary-layer wind (hereafter, $\mathbf{V}_E$) is introduced and calculated from water vapor transport scaled by the total column water vapor (i.e., $\mathbf{V}_E = \mathbf{Q}/$TPW, where TPW is the total precipitable water) to avoid the influence of water vapor trends (see Methods section). Because water vapor is mainly transported through the lower returning branch of tropical circulation (i.e., Hadley or Walker circulations), $\mathbf{V}_E$ can represent the circulation intensity and its intensity change can be interpreted as a change of the Hadley or Walker circulation intensity[36]. Detailed discussion on $\mathbf{V}_E$ is found in previous literature[37].

Figure 1a, b shows the ensemble mean distribution of velocity potential and the associated divergent wind at 200-hPa level and $\mathbf{Q}$ averaged in the control run, respectively. Note that the zonal mean total water vapor transport is removed in Fig. 1b to emphasize the zonal asymmetry caused by east–west circulations. The associated precipitation rates are given in Fig. 1c. It is found that the spatial structures of velocity potential along with 200-hPa divergent wind, $\mathbf{Q}$, and the precipitation rate obtained from the ensemble mean of 21 CMIP5 preindustrial runs are not much different from those in the control run (Supplementary Fig. 1). This similarity suggests that climate system is not meaningfully perturbed by the additional $CO_2$ in the control run (i.e., the first 20-year period); therefore, the difference between the two periods (1–20 years vs. 121–140 years) is considered representative of the atmospheric response to $QCO_2$.

The well-known Walker circulation is clearly depicted across the tropical Pacific basin. Water vapor convergence and upper-level divergence over the western tropical Pacific indicate the ascending region, connecting to the eastern tropical Pacific descending region where water vapor divergence is clearly shown. It is of importance to note that the 200-hPa level velocity potential shows a dominant wave number 1 pattern. In addition, it is also evident that the local north–south Hadley circulation is prevalent over the Indo-western tropical Pacific region. The geographical distribution of the ensemble model mean precipitation (Fig. 1c) agrees reasonably well with the expected precipitation from the water vapor convergence/divergence

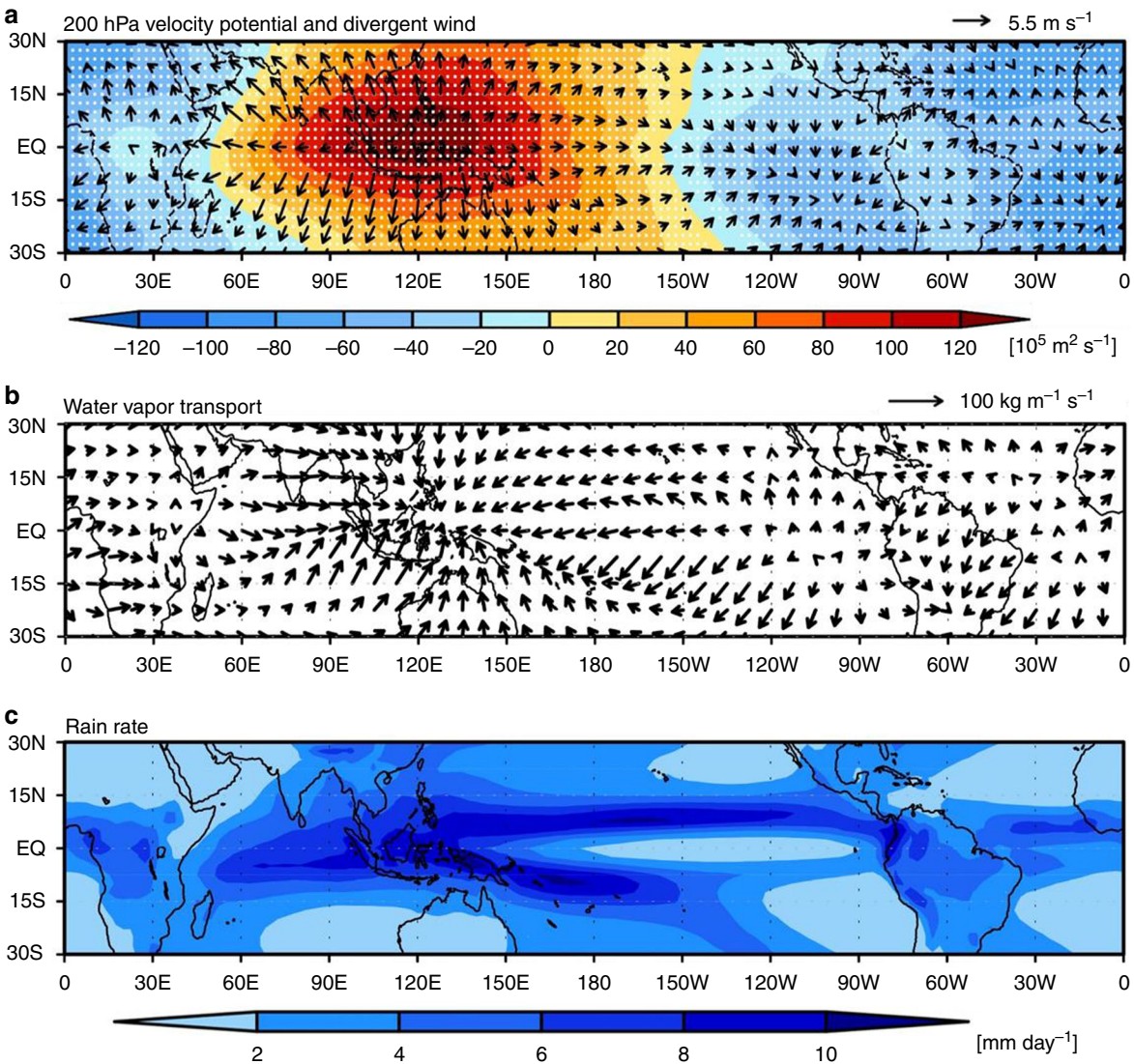

**Fig. 1** Mass overturning circulation and precipitation in the tropics. The first 20-year ensemble mean distribution of (**a**) velocity potential function (colors) and divergent wind (arrow) at 200-hPa level, (**b**) divergent water vapor transport [$Q_D$], and (**c**) rain rate in the control run. Note that the zonal mean water vapor transport is removed in **b**. Dotted denotes the region where the responses of 14 or more out of the 21 CMIP5 models are of the same sign

pattern (Fig. 1b). Heavy precipitation areas over the Indo-western Pacific region, Inter-tropical Convergence Zone (ITCZ), and South Pacific Convergence Zone (SPCZ) coincide with the water vapor convergence areas. The prevailing dry regions over the equatorial eastern Pacific and subtropical oceans coincide with water vapor divergence areas. A similar result is obtained from the ERA-Interim reanalysis dataset[33] for 1979–2017 with a wave number 1 pattern of the 200-hPa level velocity dominant in the tropics (Supplementary Fig. 2a). In addition, water vapor convergence and upper-level divergence are prominent over the Indo-western tropical Pacific along with Walker circulation and the local north–south Hadley circulation. It is also evident that much of the precipitation over the Indo-western Pacific region, ITCZ, and SPCZ are closely associated with the water vapor convergence (Supplementary Figs. 2b, c).

**Changes in precipitation and circulation to CO$_2$ increase.** In order to examine the QCO$_2$ influence on tropical precipitation, the precipitation differences between the two ensemble time means (1–20 years vs. 121–140 years) are displayed in Fig. 2a. Note that contour lines in Fig. 2a–c indicate the ensemble mean

precipitation in the control run, which is identical to Fig. 1c and the precipitation response to QCO$_2$ is highly consistent among ensemble members (Fig. 2a), which are not due to one or two models driving the maximum in precipitation change. Furthermore, it is indicated that QCO$_2$ results are more consistent among climate models, compared with the other two experiments. It is also noteworthy that the precipitation differences between the ensemble mean of 21 CMIP5 preindustrial runs and the response to QCO$_2$ (121–140 years) (Supplementary Fig. 3) are similar to that from the first to last 20 years in the quadrupling experiment of CO$_2$ (Fig. 2a).

A pronounced increase in precipitation due to QCO$_2$ is found over the western equatorial Pacific with a maximum increase over the area between 150° E and 160° W. Most of this increased precipitation area nearly corresponds to the local minimum of climatological precipitation area from 150° E to 80° W in the equatorial Pacific, which appears to be in contrast to a wet-get-wetter pattern. On the other hand, the rainfall amount is reduced in the central-to-eastern subtropical Pacific where the climatological precipitation is a minimum, which is consistent with a dry-get-drier pattern in a warmer climate[16,17]. Therefore,

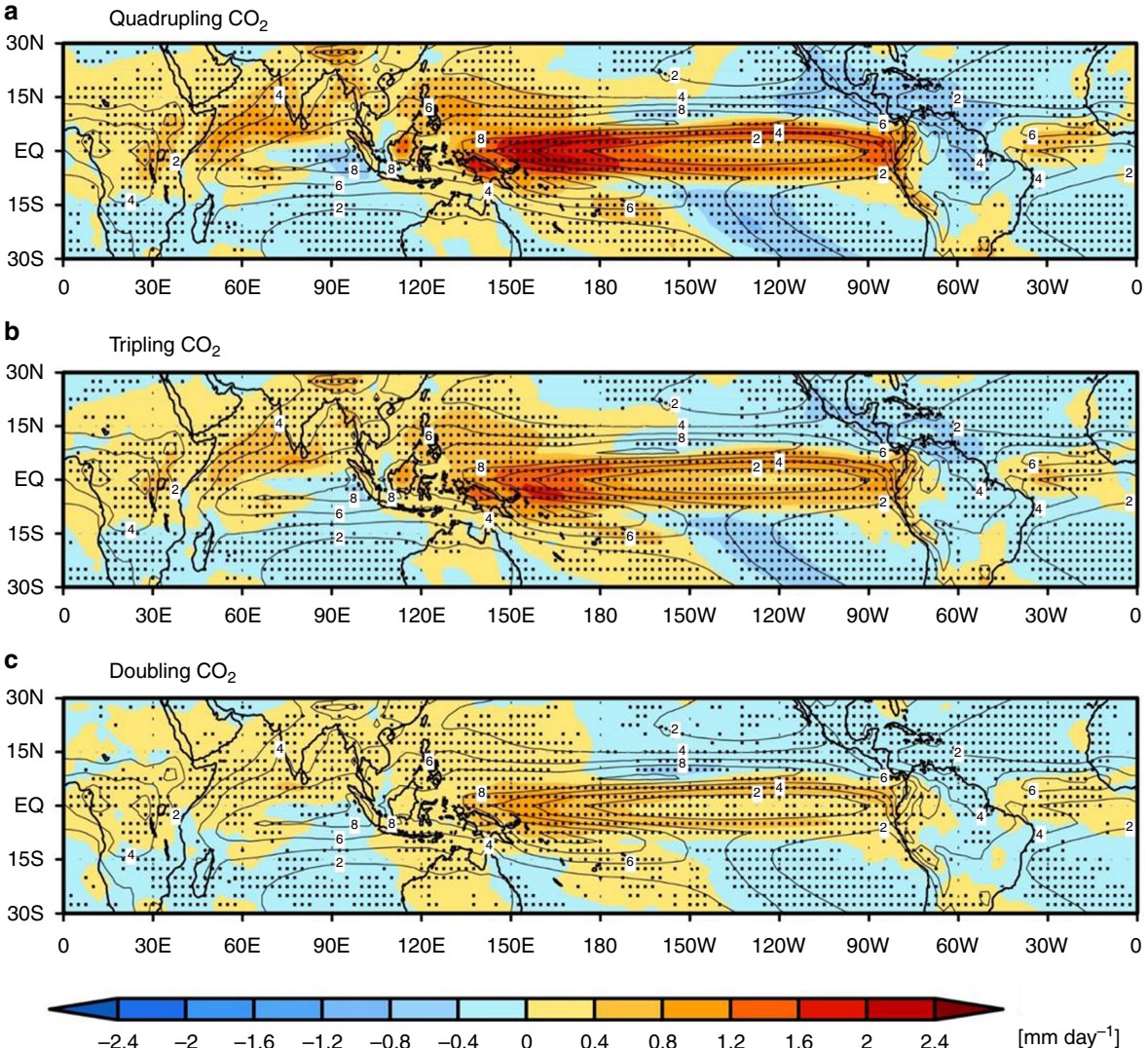

**Fig. 2** Changes in the spatial pattern of precipitation in response to $CO_2$ concentration increases. Difference distribution of ensemble mean annual precipitation for the **a** quadrupling, **b** tripling, and **c** doubling of $CO_2$ concentration. Contours represent ensemble mean field of precipitation for the first 20 years with intervals of 2 mm day$^{-1}$ in each experiment. Dotted denotes the region where the responses of 14 or more out of the 21 CMIP5 models are of the same sign

the precipitation change due to $QCO_2$ is not fully explained by a thermodynamic process. Furthermore, the details of precipitation changes have both been somewhat contrasting and consistent, when the warmer-get-wetter mechanism is employed. While both the SST warming and the SST increase relative to the tropical mean SST from the first to the last 20 years is a maximum in the eastern equatorial Pacific (Fig. 3), a maximum increase of precipitation is found over the western equatorial Pacific (Fig. 2a). On the other hand, both the western Indian Ocean and the tropical Atlantic Ocean, where the SST warming relative to the tropical mean SST is significant (Fig. 3b), are characterized by a significant increase of precipitation amount due to $QCO_2$ (Fig. 2a). This could be explained by a warmer-to-wetter mechanism. These results require a further understanding to explain the precipitation change in a warmer climate.

It should be noted that using CMIP5 model simulations, the geographical distribution of the CMIP5 model ensemble mean response in the total precipitation for doubling and tripling of $CO_2$ concentrations[11] is quite similar to that due to $QCO_2$ (Fig. 2a–c). This result indicates that the spatial structures of tropical rainfall change are not much influenced by the

magnitude of atmospheric $CO_2$ concentration change, implying that a similar mechanism may work for the doubling, tripling, and quadrupling of $CO_2$ concentrations in a warmer world. However, it is noted that the degree of inter-model agreement is higher for the $QCO_2$ experiment.

The change in the precipitation rate over the tropical Pacific to $QCO_2$ (Fig. 2a), i.e., greater increases in the western equatorial Pacific as compared with the central-to-eastern tropical Pacific, may be caused by an increased intensity of Walker circulation with a more zonally expanded pattern. This seems contradictory to the expected weakening of the Walker circulation under the global warming conditions found in most coupled climate models[36–41]. Using a typical index based on the sea-level pressure (SLP) difference between the ascending and descending regions[38,39] (see Methods section), it is obvious that the intensity of the Walker circulation to $QCO_2$ gradually decreases (Supplementary Fig. 4). This result raises a question, i.e., how can weakening of the Walker circulation explain the change in the precipitation rate of the tropical Pacific to $QCO_2$? The answer might be that the pressure changes defining the Walker circulation are not only influenced by east–west-coupled Walker

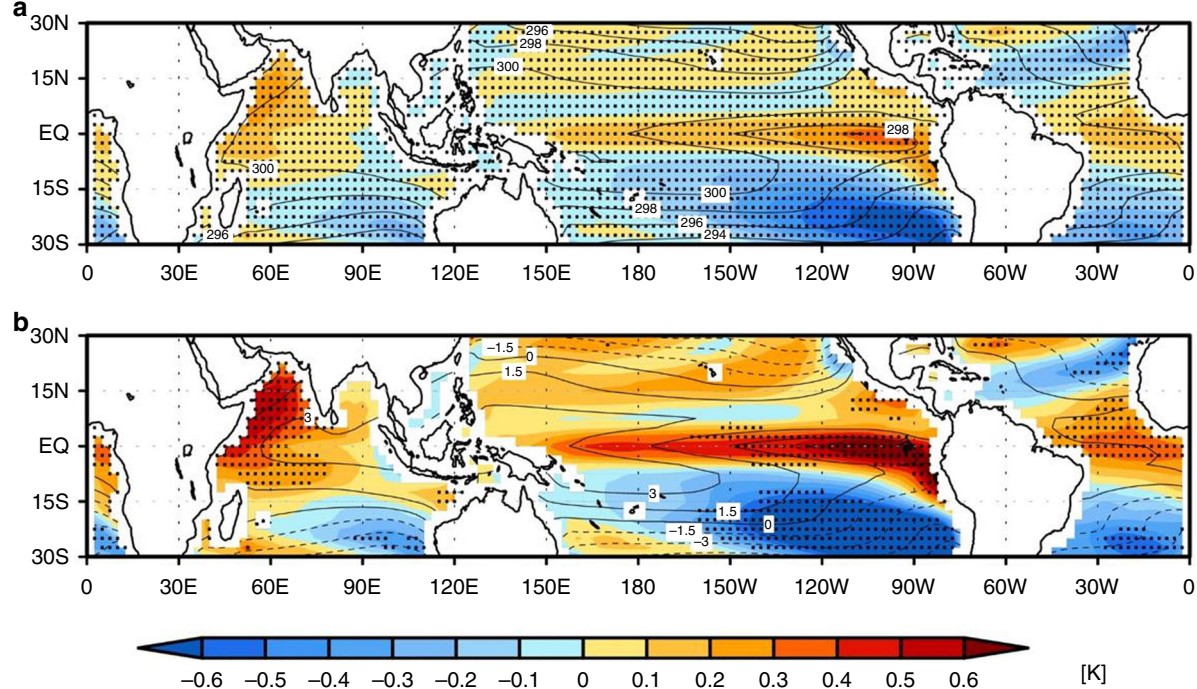

**Fig. 3** Changes in the spatial pattern of SST in response to $CO_2$ concentration increases. Difference of ensemble mean (**a**) SST between the first (1–20 years) and the last (121–140 years) 20 years in the quadrupling experiment from 21 CMIP5 models (the last 20 years minus the first 20 years). (**b**) is the same as in **a** except that the SST anomaly is obtained by subtracting the tropical (30° N–30° S) mean SST. Contours show ensemble mean SST and SST anomaly field for the first 20 years with intervals of 2 K and 1.5 K, respectively. Dotted denotes the region where the responses of 14 or more out of the 21 CMIP5 models are of the same sign

circulation but also by the north–south local Hadley circulation. Therefore, a weakening of the Walker circulation in a warmer climate, which is based on a typical index using the east–west difference of SLP, might be oversimplified and there may be changes in the structure of Walker circulation due to both thermodynamics and dynamical responses, affecting rainfall change.

In this study, the details of atmospheric circulation change are examined based on the change in the mass overturning circulation to $QCO_2$ to interpret the change in the spatial pattern of rainfall (Fig. 4). The dominant wave number 1 pattern in 200-hPa velocity potential in the control run (contour in Fig. 4a) tends to be shifted to the east, as noted in the difference field of 200-hPa divergent wind and velocity potential (shading in Fig. 4a). While the wave number 1 pattern in velocity potential at 200-hPa is still dominant in the period of $QCO_2$ (i.e., 121–140 years) (Supplementary Fig. 5), a wavenumber 1.5–2 perturbation field of velocity potential is established (see also Supplementary Fig. 6). The oval-type minimum anomalies centered over Malaysia indicate an upper-level mass convergence over the area covering the eastern part of the Indian Ocean and the Maritime continent (Fig. 4a), which is connected with an upper-level mass divergence over the tropical Atlantic Ocean and a far western part of the Indian Ocean where the SST warming relative to the tropical mean SST is significant with an increase of precipitation amount, in response to $QCO_2$ (Fig. 3b). The strongest decrease in the velocity potential roughly corresponds to the area showing the decreased precipitation over the western part of the Maritime continent and the eastern Indian Ocean (see Fig. 2a). We argue that the enhanced precipitation in the far western Indian Ocean and the tropical Atlantic Ocean (Fig. 2a), which could be explained by the warmer-get-wetter mechanism, acts to shift the

convergence/divergence structure of Walker circulation to the east in a warmer climate. On the other hand, the much elongated pattern of positive anomalies extends from the central tropical Pacific to the eastern tropical Pacific over 10° N–10° S and appears to be dynamically connected to the negative anomalies located in the west (Fig. 4a). This dipole-like distribution may suggest a link between the two regions, establishing the upper branch of the Pacific Walker circulation. However, considering that perturbed velocity is mostly linked in the north–south direction over most of the central and eastern tropical Pacific, the equatorial area extending from the central tropical Pacific to the eastern tropical Pacific is subject to more of a north–south aligned outflow from the latitudinal belt along the equator.

It is noteworthy that there is a study pointing out the eastward shift of Walker circulation represented by the zonal stream function in response to global warming, which is mainly associated with a shift toward more El Niño-like mean state due to a long-term trend in ENSO variability pattern[29]. While such an El Niño-like mean state change is also seen in the period of $QCO_2$ (Fig. 3a), the spatial structure of the mass overturning circulation associated with El Niño events has both similarities and differences (Supplementary Fig. 7a) compared with that due to $QCO_2$ (i.e., Fig. 4a). During El Niño events (see Methods section), the minimum anomalies of mass overturning circulation are located over the eastern part of the Indian Ocean and the Maritime continent, which is similar to the result in Fig. 4a. However, the elongated pattern of positive anomalies extending from the central to the eastern tropical Pacific along with the upper-level mass divergence over the tropical Atlantic Ocean and a far western Indian Ocean does not appear during El Niño events, which is in contrast to that due to $QCO_2$ (Fig. 4a). Furthermore, it is also noteworthy that the mean state change of

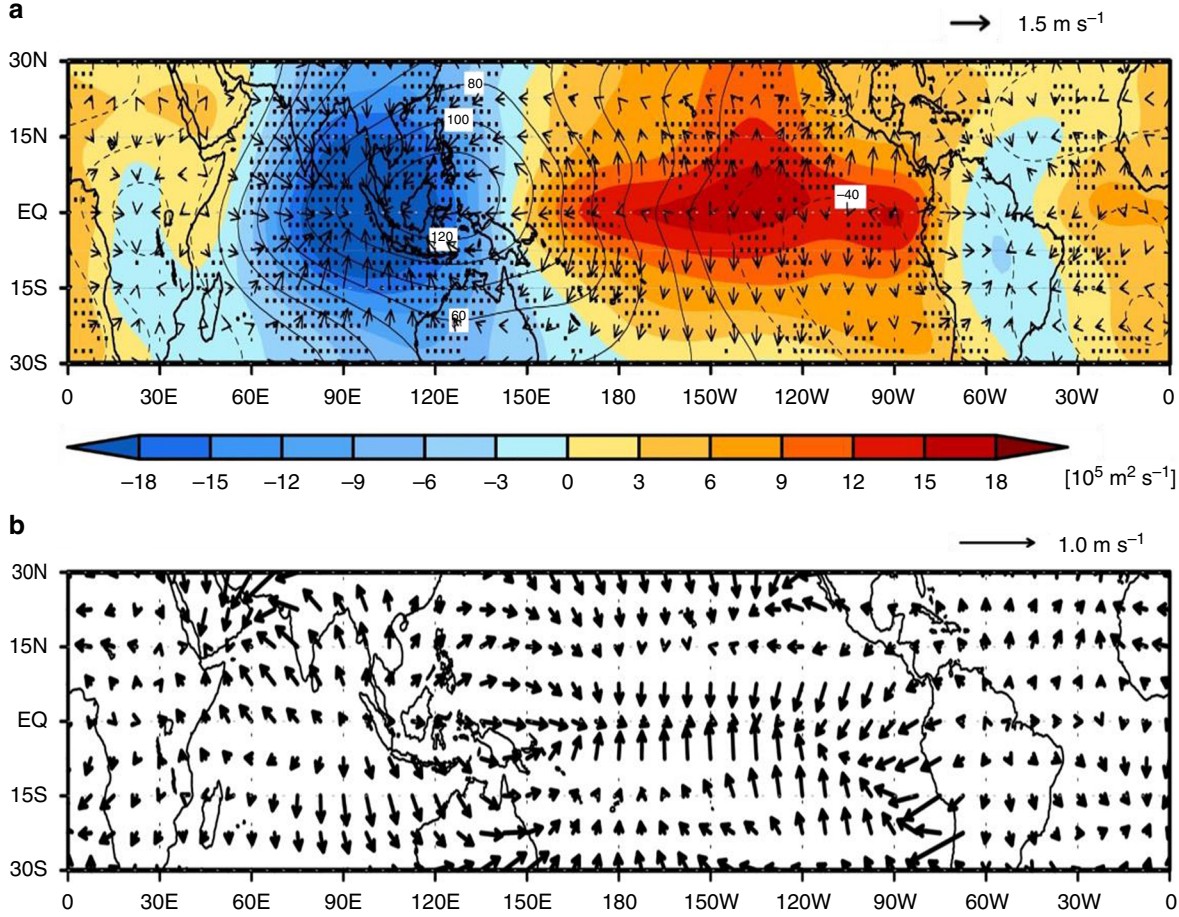

**Fig. 4** Changes in atmospheric circulation to $CO_2$ concentration increases. Difference distribution (121–140 years minus 1–20 years) of ensemble mean of (**a**) velocity potential function (colors) and divergent wind (arrow) at 200-hPa level and (**b**) effective wind [$V_E$]. Contours in **a** indicate ensemble mean field of velocity potential for the first 20 years with intervals of $20 \times 10^5$ m$^2$ s$^{-1}$. Dotted denotes the region where the responses of 14 or more out of the 21 CMIP5 models are of the same sign

SST due to $QCO_2$ (Fig. 3a) is somewhat different from the spatial pattern of El Niño events simulated in CMIP5 climate models (Supplementary Fig. 7b), in particular over the western Indian ocean and equatorial Atlantic.

The change in $V_E$ induced by $QCO_2$ is presented in Fig. 4b. In accordance with the upper-level divergence/convergence distribution shown in the 200-hPa divergent wind field (Fig. 4a), $V_E$ shows divergence over the eastern part of the Indian Ocean and convergence across the Pacific from 150° E eastward. The divergence and convergence of $V_E$ imply water vapor divergence and convergence, respectively, which explains the precipitation change to $QCO_2$ shown in Fig. 2. The increase in the velocity potential roughly corresponds to the increased rainfall amount area between 150° E and 160° W as well as the coast of South America around 90° W (Fig. 2a and Fig. 3a). In addition, such an elongated pattern across the tropical Pacific Ocean indicates that the upper-level mass fluxes diverge from the equatorial area toward the subtropical oceans in both hemispheres. This reflects a narrowing of the ITCZ, the enhanced drying of the subtropics, and marginal convective zones of the tropics[31](see also Supplementary Fig. 8), describing the change structure of the Hadley circulation due to $QCO_2$[29,30,32,33]. Therefore, the tropical circulation anomalies inferred from 200-hPa divergent wind and $V_E$ with precipitation anomalies suggest how precipitation anomalies are formed in response to $QCO_2$: precipitation over the western tropical Pacific is caused by eastward water vapor transport from the Indian Ocean, while precipitation over the

central-to-eastern tropical Pacific Ocean is largely due to the equatorward transport of water vapor from subtropical oceans (Supplementary Fig. 9). This result indicates that the changes in precipitation to $QCO_2$ might be linked to regionally confined circulations, i.e., Walker and Hadley circulations. In other words, more regionally developed circulations over the tropical Pacific basin tend to break the trans-Pacific Walker circulation into rather smaller regional scales, leading to greater increases in precipitation in the western equatorial Pacific as compared with the eastern tropical Pacific.

## Discussion
We conclude that the details of precipitation changes in response to $QCO_2$ are not directly linked by large-scale east–west-coupling across the tropical Pacific basin. By calculating water vapor transport ($Q$) and the effective boundary-layer wind ($V_E$), we proposed that regionally different responses of the Walker and Hadley circulations to increasing $CO_2$ tend to shape the details of the spatial pattern of precipitation in the tropical Pacific. Therefore, regional scales of atmospheric circulation over the tropical Pacific basin should be understood to correctly project the precipitation changes in response to increasing $CO_2$ in a warmer climate. Precipitation changes over the western equatorial Pacific seem to be more related to the Indian Ocean and the tropical Atlantic Ocean, when the eastward shift of the Walker circulation is considered. On the other hand, most central-to-eastern

equatorial Pacific regions depict an increase of local Hadley-type circulation in a warmer climate, suggesting that the conventional east–west Walker circulation in the same regions is less sensitive to $CO_2$ increases. In other words, the change in total precipitable water in the central-to-eastern tropical Pacific is largely controlled by the local Hadley circulation, consistent with intensified subtropics and a narrowing of ITCZ under the global warming condition. This somewhat contradicts the previous literature, which emphasizes the respective role of Walker circulation and Hadley circulation change in a warmer climate. In the Darwin area, for example, the change in the atmospheric circulation and the total precipitable water is greatly linked to the Indian Ocean where divergence can be expected, whereas in Tahiti, the atmospheric circulations leading to the change in the total precipitable water are nearly in the north–south direction (Supplementary Fig. 10). Thus, in many previous studies[38–41], the difference of SLP between two regions may not be interpreted as a Walker circulation change. The present study emphasizes the regulation of regional atmospheric circulation controlling the precipitation changes in response to QCO2 in a warmer climate.

## Methods

**Total water vapor transport (Q).** Because water vapor storage in the atmosphere is small over monthly timescales for a given location in comparison with evaporation (E) minus precipitation (P), water vapor excess (E − P) should be balanced by the divergence water vapor transported into the surrounding regions (**Q**), i.e., $\nabla \cdot \mathbf{Q} = E - P$. By separating the water vapor transport vector (**Q**) into rotational ($\mathbf{Q}_R$) and divergent ($\mathbf{Q}_D$) components (i.e., $\mathbf{Q} = \mathbf{Q}_R + \mathbf{Q}_D$), and by introducing the potential function of water vapor transport ($\Phi$), $\mathbf{Q}_D$ can be calculated from the relationship of $\nabla \cdot \mathbf{Q}_D = -\nabla^2 \Phi = E - P$[38,39,42]. A detailed solving method is found in a previous study[39]. For the model output, the water vapor fluxes are vertically integrated using moisture and wind profiles. After this vertical integration, only the divergent component of water vapor transport is taken[43–45]. Detailed descriptions of the calculation methods, including validations, are found in a previous study[30]. In this approach, the specific humidity (q) and horizontal wind fields (**V**) are needed to calculate water vapor flux.

The total water vapor transport (**Q**) can be defined by

$$\mathbf{Q} = -\frac{1}{g}\int_{P_s}^{P_o} q\mathbf{V}\mathrm{d}p \tag{1}$$

where g, q, **V**, $P_s$, and $P_o$ are the acceleration of gravity, specific humidity, horizontal wind vector, surface pressure, and top-of-the-atmosphere pressure, respectively.

**Effective boundary-layer wind field ($V_E$).** As expressed in equation (1), water vapor transport varies with the combination of water vapor and wind speed. Thus, both of these factors contribute to trends in water vapor flux. In other words, water vapor flux can increase under a static velocity field when the amount of water vapor increases. To understand trends in circulation strength using water vapor flux, it would be ideal if we could remove the influence of water vapor increases or decreases on the vapor transport trend. To achieve this, we introduce the concept of "effective wind ($V_E$) for water vapor transport" by considering the divergent component only, i.e.,

$$\mathbf{Q}_D = \mathrm{TPW}\sum_{i=1}^{N}\frac{\mathrm{PW}(i)}{\mathrm{TPW}}\mathbf{V}_D(i) \tag{2}$$

where TPW and PW are total precipitable water and precipitable water, respectively. Then the satellite estimate for the effective wind is given by

$$\mathbf{V}_E = \mathbf{Q}_D / \mathrm{TPW}. \tag{3}$$

On the other hand, the effective wind from model output data is from

$$\mathbf{V}_E = \sum_{i=1}^{N} W(i)\mathbf{V}_D(i). \tag{4}$$

In Eq. (4), the atmospheric column is divided into N layers, with PW(i) representing precipitable water at the ith layer. The term W(i) is the contribution of the water vapor amount by the ith layer to the TPW.

**Walker circulation index (WCI).** The Walker circulation index is defined as the pressure difference between the eastern (160° W–80° W and 5° N–5° S) and western Pacific (80° E–160° E and 5° N–5° S).

**Doubling and tripling experiments of $CO_2$ concentration.** We use outputs from 21 CMIP5 model simulations based on a 70-year and 111-year experiment with a prescribed 1% per year increase in $CO_2$ concentration, respectively, in the doubling and tripling experiment of $CO_2$ concentration. Similar to the quadrupling experiment, we consider the first 20-year (1–20 years) simulations as a control run in both the doubling and tripling experiment. Since an annual 1% increase of $CO_2$ concentration results in approximately a doubling (tripling) of $CO_2$ concentrations in the last 20 years of the 70-year (111-year) analysis period, i.e., 51–70 years (i.e., 92–111 years), the difference between the two periods, i.e., 1–20 years vs. 51–70 years (1–20 years vs. 92–111 years) is considered to be the precipitation response to doubling (tripling) of $CO_2$ concentration.

**El Niño composites in 21 CMIP5 climate models.** We counted the El Niño event as the year when the Niño 3.4 index during boreal winter (December–January–February, DJF) is above 0.4 °C during the first 20 years in each of the 21 CMIP5 climate models. In order to identify the characteristics of atmospheric states when El Niño occurs, the climatological DJF mean during the entire first 20-year period was subtracted from the years when El Niño occurs.

**Code availability.** All the IDL and Fortran 90 codes used to generate the results of this study are available from the authors upon request.

## Data availability

All CMIP5 model data are publicly available from https://esgf-node.llnl.gov/search/cmip5/.

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

## Acknowledgements

B.-J.S. and A.L. were supported by the Korea Meteorological Administration Research and Development Program under Grant KMIPA KMI2018–06910. S.-W.Y. was supported by National Research Foundation Grant NRF-2018R1A5A1024958.

## Author contributions

B.-J.S. and S.-W.Y. conceived the study, A.L. conducted analysis, and B.-J.S. and S.-W.Y. wrote the paper with comments and input from W.K.M.L.

## Additional information

**Competing interests:** The authors declare no competing interests.

