## [Peer Review File · Nature Communications]

Reviewer #1 (Remarks to the Author):

Review of "Regulation of atmospheric circulation controlling the tropical Pacific precipitation change to CO₂ increases in a warmer climate" by Sohn et al.

This manuscript attempts to explain the spatial pattern of precipitation in the tropical Pacific due to a quadrupling of CO₂. They find regional changes in the Walker and Hadley circulations that are different from previous interpretations. This result is likely worthy of publication, but the authors present an incomplete picture and unsatisfying explanation. Major revisions are required.

1. The authors relate the two prevailing notions of precipitation change with global warming: the wet-get-wetter/dry-get-drier (rich-get-richer/poor-get-poorer) pattern and the warm-get-wetter pattern. While the change in precipitation from the early to the late periods of the CMIP5 model run does partially counter the wet-get-wetter pattern (lines 137-139), there is no indication if the reason for the resultant pattern is due to a differential heating of the Pacific Ocean (warm-get-wetter). I feel that the ensemble SST difference map should be included as part of the analysis (at least as a supplementary figure).

2. The differences in the surface and upper-air fields (Figs. 2 and 3) are reminiscent of an El Niño pattern (which is driven by SST, see point 1). How do the QCO₂ changes compare to changes induced by El Niño? Would these results suggest a more frequent or permanent El Niño state? While El Niño is briefly mentioned in the introduction, much more could be surmised from the literature and included in the discussion.

3. How well do the ensemble members agree in the fields presented here? For example, Lau et al. (2013) only show regions that have high consistency among members or responses where 10 or more of the models have the same sign. Are one or two models driving the maximum in precipitation change seen between 150E and 180?

4. I would also prefer to see the actual fields of precipitation, velocity potential, divergent wind, and effective wind for the 121-140 year ensemble mean (with an indication of consistency as recommended in point 3). How does the wave number 1 pattern change in geographic position? The authors mention a wave number 2 perturbation field (line 164), but I assume the actual velocity potential field is still wave number 1.

5. The writing is uneven and I suggest some grammatical changes below. There is also a section (lines 186-193) that basically repeats earlier statements (lines 165-170).

Minor comments

1. Line 50: In addition to Cai et al. (2014), I would think Curtis and Adler (2000) would be a good reference here.

a. Curtis, S., and R. Adler, 2000: ENSO indices based on patterns of satellite-derived precipitation. *J. Climate*, 13, 2786-2793.

2. Line 62: Does it dynamically interpret or thermodynamically interpret?

3. Line 71: I suggest "which is understudied"

4. Line 77: I understand what is meant by "more increases" and "less increases", but it is awkward. What about a "greater increase in precipitation in the western tropical Pacific as compared to the eastern tropical Pacific". This phrasing also appears in the abstract.

5. Line 109: Need to mention here, as in the caption of Figure 1, that the zonal mean water vapor transport was removed.

6. Lines 137-138: I would recommend: "This change is somewhat in contrast..."

7. Line 167: "anomalies" instead of "area"

8. Line 168: "appears to be dynamically connected to" instead of "appears to pair with"

9. Lines 183-184: There is another center of high velocity potential difference (the absolute maximum) and high precipitation difference off the coast of South America around 90W. Very little is said about this area. Also, it should be Fig. 3a and not Fig. 3b.

10. Line 184: Add "an" after "Such"

11. References: The citation format varies greatly.

12. Supplementary Figure 5. Why is TPW shown? I don't see what information it adds.

Scott Curtis

Reviewer #2 (Remarks to the Author):

The authors present a study of the changes in tropical Pacific precipitation patterns and circulation as a response to CO₂ concentration quadrupling. They use the outputs of 20 GCMs for the 1pctCO₂ idealized CMIP5 climate experiment. For quantifying the circulation and precipitation intensities they make use of methods already well-represented in literature. The authors argue that the precipitation changes in this region are caused by the combined action of the response of the Walker and the Hadley circulations to CO₂ forcing. Moreover, they suggest a dominating role of the Walker circulation change on the Maritime continent region and west Pacific and of the Hadley circulation change on the central and eastern Pacific.

Recommendation: Accept with major revisions

The manuscript is well structured and the authors have constructed a logic line of argument to prove their theory. However, although approaching the investigation of the change in precipitation from this particular point of view is reasonable, there are several issues I would suggest to be addressed before accepting the manuscript for publication. Please see below the comments, the addressing of which I hope will improve the manuscript.

Major comments:

1. The authors propose the combination of the changes in Walker and Hadley circulation as the mechanism driving the pattern of precipitation changes over the tropical Pacific. However, this is not the first study showing the particular pattern of divergence/convergence fields across the equatorial Indian-Pacific basin (see Bayr et al. 2014). There is also not sufficient discussion of the mechanisms proposed by other previous studies, e.g. the location of the ascending branch of the Walker circulation (and the associated convection region) or the narrowing of the ITCZ. The manuscript would benefit from a more thorough discussion of these and a much clearer defined novelty of the findings. Please see a list of suggested papers at the end of the comments.

2. The results presented in the manuscript are presented solely from the perspective of the CMIP5 ensemble mean. There is, however, a known large spread in the climate system response detected in the various models. The authors do not show any statistical analysis related to their results to back up the significance of the claims. Such investigation would be desirable at least (!) as supplementary material.

3. Please elaborate on the choice of the first 20 years of the 1pctCO₂ runs as a control state. As in this period the climate system is already responding to the additional CO₂, why not use a mean over the piControl runs as a control state?

Minor comments:

1. Lines 54, 213: "... CO₂ increases in a warmer climate": this study does not treat the carbon cycle, therefore such a phrasing is misleading, as the 1pctCO₂ experiment treats the warmer climate as a result of CO₂ increase. Please reformulate.

2. Line 71: "... CMIP5 CO₂ experiment results...": there are several idealized climate experiments, please state which one is referred here.

3. Lines 76-77: "... more increases in the western tropical Pacific and less increases in the eastern tropical Pacific": please reformulate to explain the increase of what parameter is referred here. Also, please explain what do you mean by "more increases": more regions of increase or a stronger rate of increase?

4. Line 80: "CO2 emissions": 1pctCO2 is not an emission scenario, suggested reformulation to "CO2 concentration".
5. Lines 81-82: "the impact of major anthropogenic GHGs": in the previous sentence the authors state that only CO2 varies, therefore such a formulation is contradictory.
6. Lines 144-149: The argumentation is unclear: the authors have not investigated the individual impact of CO2 increase on the circulation/precipitation, as the climate system adjusts to the increased concentrations. Hopefully, a rephrasing would clarify this issue. Also, a strengthened Walker circulation would not necessarily be confined in the same zonal/meridional boundaries and would have a more zonally expanded pattern of increased precipitation (with more increase over the Maritime Continent). Therefore, the phrase in lines 148-149 would benefit from more clarification.
7. Lines 167-173: Please add reference to the Fig. 3, where needed.
8. Line 181: After "... across the Pacific from 150 degE" it would be clearer if "eastward" is added.
9. Lines 189-190: "... the area showing less precipitation change with negative or positive sign": What does the sign refer to?
10. Lines 353-373: Weren't these metrics introduced in previous studies?
11. The manuscript would highly benefit from a proof-read made by a native speaker.
12. The format of the references should be consistent, e.g. see referenced 17, 18, 20, as well as removal of typos, e.g. reference 35.
13. Suppl. Fig. 1: Please remove "(control)" from the titles of the subplots.
14. Suppl. Fig. 2: Either refer to the Lau et al. 2013 paper without including the figure or adapt the figure in the style of the previous ones from the manuscript.

References:

Ma, S. and Zhou, T. (2016). Robust strengthening and westward shift of the tropical Pacific Walker circulation during 1979–2012: A comparison of 7 sets of reanalysis data and 26 CMIP5 models. *Journal of Climate*, 29(9):3097–3118.

McGregor, S., Timmermann, A., Stuecker, M. F., England, M. H., Merrifield, M., Jin, F.-F., and Chikamoto, Y. (2014). Recent Walker circulation strengthening and Pacific cooling amplified by Atlantic warming. *Nature Climate Change*, 4(10):888–892.

Bayr, T., Dommenges, D., Martin, T., and Power, S. B. (2014). The eastward shift of the Walker circulation in response to global warming and its relationship to ENSO variability. *Climate Dynamics*, 43(9-10):2747–2763.

Byrne, M. P. and Schneider, T. (2016). Narrowing of the ITCZ in a warming climate: Physical mechanisms. *Geophysical Research Letters*, 43(21): 11350–11357.

Optional:

He, J. and Soden, B. J. (2015). Anthropogenic weakening of the tropical circulation: The relative roles of direct CO2 forcing and sea surface temperature change. *Journal of Climate*, 28(22):8728–8742.

Su, H., Jiang, J. H., Zhai, C., Shen, T. J., Neelin, J. D., Stephens, G. L., and Yung, Y. L. (2014). Weakening and strengthening structures in the Hadley circulation change under global warming

and implications for cloud response and climate sensitivity. *Journal of Geophysical Research: Atmospheres*, 119(10):5787–5805.

Reviewers' comments:

Reviewer #1 (Remarks to the Author):

Review of “Regulation of atmospheric circulation controlling the tropical Pacific precipitation change to CO₂ increases in a warmer climate” by Sohn et al.

This manuscript attempts to explain the spatial pattern of precipitation in the tropical Pacific due to a quadrupling of CO₂. They find regional changes in the Walker and Hadley circulations that are different from previous interpretations. This result is likely worthy of publication, but the authors present an incomplete picture and unsatisfying explanation. Major revisions are required.

:We are grateful to this reviewer which provided very valuable comments which significantly improved the earlier version of the manuscript. The manuscript has been revised according to the reviewer’s comments. In addition, we found a small mistake to calculate the velocity potential responding to the quadrupling of CO₂ concentration in the previous manuscript and it has been corrected in the revised version (Fig. 4 in the revised manuscript) although it does little change the results presented in the current study. Following the reviewer #1 and #2’s comments, the earlier version of the manuscript has been mainly revised as follows.

- We emphasized that more regionally perturbed circulations over the tropical Pacific, which is influenced by the enhanced precipitation outside the tropical Pacific, lead to greater increases in precipitation in the western equatorial Pacific as compared to the eastern tropical Pacific in a warmer climate.
- We analyzed outputs from doubling and tripling experiments of CO₂ concentration from 21 CMIP5 model simulations and 21 CMIP5 pre-industrial simulations to compare with the results obtained from a quadrupling experiment of CO₂ concentration.
- We examined whether the precipitation response and mass overturning circulation due to QCO₂ is consistent among ensemble members. The quadrupling experiment looks more credible than other two experiments since inter-model discrepancies appear to be smaller.
- We analyzed the SST warming itself and the SST warming relative to the tropical mean SST due to QCO₂ to understand the details of spatial pattern of precipitation and then compared with previous literature.

- We examined the spatial structure of the mass overturning circulation associated with El Nino events to understand the changes of Walker circulation due to QCO₂. While there is a similarity such as warming over the eastern Pacific, differences are also noted in particular over the western Indian Ocean and equatorial Atlantic.

We assert that the above-listed revisions support and strengthen our main conclusions regarding the regionally different response of the Walker and Hadley circulations to CO₂ increase and the impact in shaping precipitation across the tropical Pacific

1. The authors relate the two prevailing notions of precipitation change with global warming: the wet-get-wetter/dry-get-drier (rich-get-richer/poor-get-poorer) pattern and the warm-get-wetter pattern. While the change in precipitation from the early to the late periods of the CMIP5 model run does partially counter the wet-get-wetter pattern (lines 137-139), there is no indication if the reason for the resultant pattern is due to a differential heating of the Pacific Ocean (warm-get-wetter). I feel that the ensemble SST difference map should be included as part of the analysis (at least as a supplementary figure).

:The reviewer's comments are grateful. We appreciate it. Responding to the reviewer's we analyze the ensemble mean SST difference and add the following sentences with a new figure in the revised manuscript:

“A pronounced increase in precipitation due to QCO₂ is found over the western equatorial Pacific with a maximum increase over the area between 150°E and 160°W. Most of this increased precipitation area nearly corresponds to the local minimum of climatological precipitation area from 150°E to 80°W in the equatorial Pacific, which appears to be in contrast to a wet-get-wetter pattern. On the other hand, the rainfall amount is reduced in the central-to-eastern subtropical Pacific where the climatological precipitation is a minimum, which is consistent with a dry-get-drier pattern in a warmer climate¹⁶⁻¹⁷. Therefore, the precipitation change due to QCO₂ is not fully explained by thermodynamic process. Furthermore, the details of precipitation changes have both somewhat contrast and consistency, when the warmer-get-wetter mechanism is employed. While both the SST warming and the SST increase relative to the tropical mean SST from the first to last 20- year is a maximum in the eastern equatorial Pacific (Fig. 3), a maximum increase of precipitation is found over the western equatorial Pacific (Fig. 2a). On the other hand, both the western Indian Ocean and the tropical Atlantic Ocean, where the SST warming relative to the tropical

mean SST is significant (Fig. 3b), are characterized by a significant increase of precipitation amount due to QCO₂ (Fig. 2a). This could be explained by a warmer-to-wetter mechanism. These results require a further understanding to explain the precipitation change in a warmer climate.”

Figure 3 Difference of ensemble mean (a) SST between the first (1-20 years) and the last (121-140 years) 20 years in the quadrupling experiment from 21 CMIP5 models (the last 20 years minus the first 20 years). (b) is the same as in Fig. 3a except the SST anomaly obtained by subtracting tropical (30°N – 30°S) mean SST. Contours show ensemble mean SST and SST anomaly field for the first 20 years with intervals of 2 K and 1.5 K, respectively. Dotted denotes the region where the responses of 14 or more out of the 21 CMIP5 models are of the same sign.

2. The differences in the surface and upper-air fields (Figs. 2 and 3) are reminiscent of an El Niño pattern (which is driven by SST, see point 1). How do the QCO₂ changes compare to changes induced by El Niño? Would these results suggest a more frequent or permanent El Niño state? While El Niño is briefly mentioned in the introduction, much more could be surmised from the literature and included in the discussion.

:The reviewer’s opinion is right. Following the above comment #1, we add the following sentences with additional Supplementary figures in the revised manuscript .

“In this study, the details of atmospheric circulation change are examined based on the change in the mass overturning circulation to QCO₂ to interpret the change in spatial pattern of rainfall (Fig. 4). The dominant wave number 1 pattern in 200 hPa velocity potential in the control run (contour in Fig. 4a) tends to be shifted to the east as noted in the difference field of 200hPa divergent wind and velocity potential (shading in Fig. 4a). While the wave number 1 pattern in velocity potential at 200hPa is still dominant in the period of QCO₂ (i.e., 121-140 years) (Supplementary Fig. 5), a wavenumber 1.5~2 perturbation field of velocity potential is established (see also Supplementary Fig. 6). The oval type minimum anomalies centered over Malaysia indicate an upper-level mass convergence over the area covering the eastern part of the Indian Ocean and the Maritime continent (Fig. 4a), which is connected with an upper-level mass divergence over the tropical Atlantic Ocean and a far western part of the Indian Ocean where the SST warming relative to the tropical mean SST is significant with an increase of precipitation amount, in response to QCO₂ (Fig. 3b). The strongest decrease in the velocity potential roughly corresponds to the area showing the decreased precipitation over the western part of the Maritime continent and the eastern Indian Ocean (see Fig. 2a). We argue that the enhanced precipitation in the far western Indian Ocean and the tropical Atlantic Ocean (Fig. 2a), which could be explained by the warmer-get-wetter mechanism, acts to shift the convergence/divergence structure of Walker circulation to the east in a warmer climate. On the other hand, the much elongated pattern of positive anomalies extend from the central tropical Pacific to the eastern tropical Pacific over 10°N-10°S and appears to be dynamically connected to the negative anomalies located in the west (Fig. 4a). This dipole-like distribution may suggest a link between the two regions, establishing the upper branch of the Pacific Walker circulation. However, considering that perturbed velocity is mostly linked in the north-south direction over most of the central and eastern tropical Pacific. That is, the equatorial area extending from the central tropical Pacific to the eastern tropical Pacific is subject to more of a north-south aligned outflow from the latitudinal belt along the equator.

It is noteworthy that there is a study pointing out the eastward shift of Walker circulation represented by the zonal stream function in response to global warming, which is mainly associated with a shift toward more El Nino-like mean state due to a long-term trend in ENSO variability pattern²⁹. While such an El Nino-like mean state change is also seen in the period of QCO₂ (Fig. 3a), the spatial structure of the mass overturning circulation associated with El Nino events has both similarities and differences (Supplementary Fig. 7a) compared with that due to QCO₂ (i.e., Fig. 4a). During El Nino events (see Method sections),

the minimum anomalies of mass overturning circulation are located over the eastern part of the Indian Ocean and the Maritime continent, which is similar to the result in Fig. 4a. However, the elongated pattern of positive anomalies extending from the central to the eastern tropical Pacific along with the upper-level mass divergence over the tropical Atlantic Ocean and a far western Indian Ocean does not appear during El Nino events, which is in contrast to that due to QCO₂ (Fig. 4a). Furthermore, it is also noteworthy that the mean state change of SST due to QCO₂ (Fig. 3a) is somewhat different from the spatial pattern of El Nino events simulated in CMIP5 climate models (Supplementary Fig. 7b), in particular over the western Indian ocean and equatorial Atlantic.”

Supplementary Figure 5 Same as in Fig. 1, but for the last 20 years (121-140 years) in the quadrupling of CO₂ concentration.

Supplementary Figure 6 Difference distribution (121-140 years minus 1-20 years) of 200 hPa velocity potential averaged in the 10°N – 10°S in each 21 CMIP5 models (gray line) and their ensemble mean (red line).

Supplementary Figure 7 (a) The ensemble mean composited map of 200hPa velocity potential and divergent wind in the years when El Nino occurs during boreal winter (December-January-February) in the first 20-years from 21 CMIP5 model simulation (quadrupling experiment of CO₂ concentration). (b) is the same as in (a) except SST anomaly. Contours in (b) denotes the ensemble mean SST during boreal winter in the first 20-years

from 21 CMIP5 model simulation. Dotted in (a), (b) denotes the region where the responses of 14 or more out of the 21 CMIP5 models are of the same sign.

3. How well do the ensemble members agree in the fields presented here? For example, Lau et al. (2013) only show regions that have high consistency among members or responses where 10 or more of the models have the same sign. Are one or two models driving the maximum in precipitation change seen between 150E and 180?

:Thank you for the reviewer's careful comment. Responding to the reviewer's comment, we examined how well do the ensemble members agree in the fields presented in the current study (please see Figs. 1-4 in the revised manuscript and additional supplementary figures). Responding to the reviewer's comments, we add the following sentences in the revised manuscript:

“Note that contour lines in Fig. 2a-c indicate the ensemble mean precipitation in the control run, which is identical to Fig. 1c and the precipitation response to QCO₂ is highly consistent among ensemble members (Fig 2a), which are not due to one or two models driving the maximum in precipitation change. Furthermore, it is indicated that QCO₂ results are more consistent among climate models, compared to other two experiments. It is also noteworthy that the precipitation differences between the ensemble mean of 21 CMIP5 pre-industrial runs and the response to QCO₂ (121-140 years) (Supplementary Fig. 3) are similar to that from the first to last 20- years in the quadrupling experiment of CO₂ (Fig. 2a). ”

Following the reviewer's comments, the figure captions are revised as follows:

Figure 1. First 20-year ensemble mean distribution of (a) velocity potential function (colors) and divergent wind (arrow) at 200 hPa level, (b) divergent water vapor transport [Q_D], and (c) rain rate in the control run. Note that the zonal mean water vapor transport is removed in Fig. 1b. Dotted denotes the region where the responses of 14 or more out of the 21 CMIP5 models are of the same sign.

Figure 2. Difference distribution of ensemble mean annual precipitation for the (a) quadrupling, (b) tripling and (c) doubling of CO₂ concentration. Contours represent ensemble mean field of precipitation for the first 20 years with intervals of 2 mm day⁻¹ in each

experiment. Dotted denotes the region where the responses of 14 or more out of the 21 CMIP5 models are of the same sign.

Figure 3 Difference of ensemble mean (a) SST between the first (1-20 years) and the last (121-140 years) 20 years in the quadrupling experiment from 21 CMIP5 models (the last 20 years minus the first 20 years). (b) is the same as in Fig. 3a except the SST anomaly obtained by subtracting tropical (30°N – 30°S) mean SST. Contours show ensemble mean SST and SST anomaly field for the first 20 years with intervals of 2 K and 1.5 K, respectively. Dotted denotes the region where the responses of 14 or more out of the 21 CMIP5 models are of the same sign.

Figure 4. Difference distribution (121-140 years minus 1-20 years) of ensemble mean of (a) velocity potential function (colors) and divergent wind (arrow) at 200 hPa level and (b) effective wind [V_E]. Contours in (a) indicate ensemble mean field of velocity potential for the first 20 years with intervals of $20 \times 10^5 \text{ m}^2 \text{ s}^{-1}$. Dotted denotes the region where the responses of 14 or more out of the 21 CMIP5 models are of the same sign.

4. I would also prefer to see the actual fields of precipitation, velocity potential, divergent wind, and effective wind for the 121-140 year ensemble mean (with an indication of consistency as recommended in point 3). How does the wave number 1 pattern change in geographic position? The authors mention a wave number 2 perturbation field (line 164), but I assume the actual velocity potential field is still wave number 1.

:Thank you for the reviewer’s careful comment. The reviewer’s comment is right. Following the reviewer’s comment, we provide the actual fields of precipitation, velocity potential, divergent and effective wind for the 121-140 year ensemble mean with an indication of consistency. (Supplementary Figs. 5, 6 in the revised manuscript). Responding to the reviewer’s comment, the following sentences are added in the revised manuscript.

“In this study, the details of atmospheric circulation change are examined based on the change in the mass overturning circulation to QCO₂ to interpret the change in spatial pattern of rainfall (Fig. 4). The dominant wave number 1 pattern in 200 hPa velocity potential in the control run (contour in Fig. 4a) tends to be shifted to the east as noted in the difference field of 200hPa divergent wind and velocity potential (shading in Fig. 4a). While the wave number 1 pattern in velocity potential at 200hPa is still dominant in the period of QCO₂ (i.e., 121-140

years) (Supplementary Fig. 5), a wavenumber 1.5~2 perturbation field of velocity potential is established (see also Supplementary Fig. 6).”

5. The writing is uneven and I suggest some grammatical changes below. There is also a section (lines 186-193) that basically repeats earlier statements (lines 165-170).

:Thank you for the reviewer’s careful comments. We appreciate it. Responding to the reviewer’s comments, we carefully read the earlier version of the manuscript and improved their writing.

Minor comments

1. Line 50: In addition to Cai et al. (2014), I would think Curtis and Adler (2000) would be a good reference here

a. Curtis, S., and R. Adler, 2000: ENSO indices based on patterns of satellite-derived precipitation. J. Climate, 13, 2786-2793.

: Added.

2. Line 62: Does it dynamically interpret or thermodynamically interpret?

: Corrected.

3. Line 71: I suggest “which is understudied”

: Corrected.

4. Line 77: I understand what is meant by “more increases” and “less increases”, but it is awkward. What about a “greater increase in precipitation in the western tropical Pacific as compared to the eastern tropical Pacific”. This phrasing also appears in the abstract.

:We corrected as follows:

Abstract:

“lead to greater increases in precipitation in the western equatorial Pacific as compared to the eastern tropical Pacific in a warmer climate.”

“In this study, using a single metric of water vapor transport in the tropics, we demonstrate that the CO₂ increase tends to break up the trans-Pacific Walker circulation into more regionally confined circulations in smaller scales, resulting in greater increases in precipitation over the western tropical Pacific as compared to the eastern tropical Pacific.”

5. Line 109: *Need to mention here, as in the caption of Figure 1, that the zonal mean water vapor transport was removed.*

:We added the following sentence:

“Note that the zonal mean total water vapor transport is removed in Fig. 1b to emphasize the zonal asymmetry caused by east-west circulation.”

6. Lines 137-138: *I would recommend: “This change is somewhat in contrast...”*

: Corrected.

7. Line 167: *“anomalies” instead of “area”*

:Corrected.

8. Line 168: *“appears to be dynamically connected to” instead of “appears to pair with”*

:Corrected.

9. Lines 183-184: *There is another center of high velocity potential difference (the absolute maximum) and high precipitation difference off the coast of South America around 90W. Very little is said about this area. Also, it should be Fig. 3a and not Fig. 3b.*

:We revised as follows:

“The increase in the velocity potential roughly corresponds to the increased rainfall amount area between 150°E and 160°W as well as the coast of South America around 90°W (Fig. 2a and Fig. 3a). In addition, such an elongated pattern across the tropical Pacific Ocean indicates the upper-level mass fluxes diverge from the equatorial area toward the subtropical oceans in both hemispheres. This reflects a narrowing of the ITCZ, the enhanced and drying of the subtropics, and marginal convective zones of the tropics³¹(see also Supplementary Fig. 8), describing the change structure of the Hadley circulation due to QCO₂^{29-30,32-33}.”

10. Line 184: *Add “an” after “Such”*

“Added.

11. References: *The citation format varies greatly.*

:Corrected.

12. Supplementary Figure 5. *Why is TPW shown? I don't see what information it adds.*

:The reason why we showed TPW with an effective wind in Supplementary Fig. 10 (in the revised manuscript) is because we want to show how the change in total precipitable water is associated with an east-west zonal connection of the Walker circulation and an increase of local Hadley-type circulation. Supplementary Fig. 10 shows that the change in the atmospheric circulation in the Darwin region is greatly linked to the Indian Ocean where divergence can be expected, whereas in Tahiti, the atmospheric circulations are nearly in the north-south direction.

Responding to the reviewer's comment, we revised as follows:

“In other words, the change in total precipitable water in the central-to-eastern tropical Pacific is largely controlled by the local Hadley circulation, consistent with intensified subtropics and a narrowing of ITCZ under the global warming condition. This somewhat contradicts the previous literature, which emphasize the respective role of Walker circulation and Hadley circulation change in a warmer climate. In the Darwin area, for example, the change in the atmospheric circulation and the total precipitable water is greatly linked to the Indian Ocean where divergence can be expected, whereas in Tahiti, the atmospheric circulations leading to the change in the total precipitable water are nearly in the north-south direction (Supplementary Fig. 10).”

Scott Curtis

Reviewer #2 (Remarks to the Author):

The authors present a study of the changes in tropical Pacific precipitation patterns and circulation as a response to CO₂ concentration quadrupling. They use the outputs of 20 GCMs for the 1pctCO₂ idealized CMIP5 climate experiment. For quantifying the circulation and precipitation intensities they make use of methods already well-represented in literature. The authors argue that the precipitation changes in this region are caused by the combined action of the response of the Walker and the Hadley circulations to CO₂ forcing. Moreover, they suggest a dominating role of the Walker circulation change on the Maritime continent region and west Pacific and of the Hadley circulation change on the central and eastern Pacific.

Recommendation: Accept with major revisions

The manuscript is well structured and the authors have constructed a logic line of argument to prove their theory. However, although approaching the investigation of the change in precipitation from this particular point of view is reasonable, there are several issues I would suggest to be addressed before accepting the manuscript for publication. Please see below the comments, the addressing of which I hope will improve the manuscript.

:We are grateful to this reviewer who gave very constructive comments, which improve the earlier version of the manuscript. We addressed all issues raised by the reviewer and revised the manuscript accordingly. In addition, we found a small mistake to calculate the velocity potential responding to the quadrupling of CO₂ concentration in the previous manuscript and it has been corrected in the revised version (Fig. 4 in the earlier version of the manuscript) although it does little change the results presented in the current study. Following the reviewer #1 and #2's comments, the earlier version of the manuscript has been mainly revised as follows:

- We emphasized that more regionally perturbed circulations over the tropical Pacific, which is influenced by the enhanced precipitation outside the tropical Pacific, lead to greater increases in precipitation in the western equatorial Pacific as compared to the eastern tropical Pacific in a warmer climate.
- We analyzed outputs from doubling and tripling experiments of CO₂ concentration from 21 CMIP5 model simulations and 21 CMIP5 pre-industrial simulations to compare with the results obtained from a quadrupling experiment of CO₂ concentration.
- We examined whether the precipitation response and mass overturning circulation due to

QCO₂ is consistent among ensemble members. The quadrupling experiment looks more credible than other two experiments since inter-model discrepancies appear to be smaller.

- We analyzed the SST warming itself and the SST warming relative to the tropical mean SST due to QCO₂ to understand the details of spatial pattern of precipitation and then compared with previous literature.
- We examined the spatial structure of the mass overturning circulation associated with El Nino events to understand the changes of Walker circulation due to QCO₂. While there is a similarity such as warming over the eastern Pacific, differences are also noted in particular over the western Indian Ocean and equatorial Atlantic.

We assert that the above-listed revisions support and strengthen our main conclusions regarding the regionally different response of the Walker and Hadley circulations to CO₂ increase and the impact in shaping precipitation across the tropical Pacific

Major comments:

1. The authors propose the combination of the changes in Walker and Hadley circulation as the mechanism driving the pattern of precipitation changes over the tropical Pacific. However, this is not the first study showing the particular pattern of divergence/convergence fields across the equatorial Indian-Pacific basin (see Bayr et al. 2014). There is also not sufficient discussion of the mechanisms proposed by other previous studies, e.g. the location of the ascending branch of the Walker circulation (and the associated convection region) or the narrowing of the ITCZ. The manuscript would benefit from a more thorough discussion of these and a much clearer defined novelty of the findings. Please see a list of suggested papers at the end of the comments.

:Thank you for the reviewer's valuable comments. We agree with the reviewer's opinion that it is necessary to discuss the mechanism proposed by other previous studies. Responding the reviewer's comments, the following sentences are added in the revised manuscript

“Previous studies have emphasized the oceanic regulations determining precipitation pattern. Furthermore, recent studies are paying more attention to the respective roles of Walker and Hadley circulation, determining the structure of divergence/convergence driving the pattern of precipitation changes over the tropical Pacific in a warmer climate²⁹⁻³³. In this study, using a single metric of water vapor transport in the tropics, we demonstrate that the CO₂ increase tends to break up the trans-Pacific Walker circulation into more regionally confined

circulations in smaller scales, resulting in greater increases in precipitation over the western tropical Pacific as compared to the eastern tropical Pacific. ”

“A pronounced increase in precipitation due to QCO₂ is found over the western equatorial Pacific with a maximum increase over the area between 150°E and 160°W. Most of this increased precipitation area nearly corresponds to the local minimum of climatological precipitation area from 150°E to 80°W in the equatorial Pacific, which appears to be in contrast to a wet-get-wetter pattern. On the other hand, the rainfall amount is reduced in the central-to-eastern subtropical Pacific where the climatological precipitation is a minimum, which is consistent with a dry-get-drier pattern in a warmer climate¹⁶⁻¹⁷. Therefore, the precipitation change due to QCO₂ is not fully explained by thermodynamic process. Furthermore, the details of precipitation changes have both somewhat contrast and consistency, when the warmer-get-wetter mechanism is employed. While both the SST warming and the SST increase relative to the tropical mean SST from the first to last 20- year is a maximum in the eastern equatorial Pacific (Fig. 3), a maximum increase of precipitation is found over the western equatorial Pacific (Fig. 2a). On the other hand, both the western Indian Ocean and the tropical Atlantic Ocean, where the SST warming relative to the tropical mean SST is significant (Fig. 3b), are characterized by a significant increase of precipitation amount due to QCO₂ (Fig. 2a). This could be explained by a warmer-to-wetter mechanism. These results require a further understanding to explain the precipitation change in a warmer climate.”

“It is noteworthy that there is a study pointing out the eastward shift of Walker circulation represented by the zonal stream function in response to global warming, which is mainly associated with a shift toward more El Nino-like mean state due to a long-term trend in ENSO variability pattern²⁹. While such an El Nino-like mean state change is also seen in the period of QCO₂ (Fig. 3a), the spatial structure of the mass overturning circulation associated with El Nino events has both similarities and differences (Supplementary Fig. 7a) compared with that due to QCO₂ (i.e., Fig. 4a). During El Nino events (see Method sections), the minimum anomalies of mass overturning circulation are located over the eastern part of the Indian Ocean and the Maritime continent, which is similar to the result in Fig. 4a. However, the elongated pattern of positive anomalies extending from the central to the eastern tropical Pacific along with the upper-level mass divergence over the tropical Atlantic Ocean and a far

western Indian Ocean does not appear during El Nino events, which is in contrast to that due to QCO₂ (Fig. 4a). Furthermore, it is also noteworthy that the mean state change of SST due to QCO₂ (Fig. 3a) is somewhat different from the spatial pattern of El Nino events simulated in CMIP5 climate models (Supplementary Fig. 7b), in particular over the western Indian ocean and equatorial Atlantic.”

“The increase in the velocity potential roughly corresponds to the increased rainfall amount area between 150°E and 160°W as well as the coast of South America around 90°W (Fig. 2a and Fig. 3a). In addition, such an elongated pattern across the tropical Pacific Ocean indicates the upper-level mass fluxes diverge from the equatorial area toward the subtropical oceans in both hemispheres. This reflects a narrowing of the ITCZ, the enhanced and drying of the subtropics, and marginal convective zones of the tropics³¹(see also Supplementary Fig. 8), describing the change structure of the Hadley circulation due to QCO₂^{29-30,32-33}.”

Figure 2. Difference distribution of ensemble mean annual precipitation for the (a) quadrupling, (b) tripling and (c) doubling of CO₂ concentration. Contours represent ensemble mean field of precipitation for the first 20 years with intervals of 2 mm day⁻¹ in each experiment. Dotted denotes the region where the responses of 14 or more out of the 21 CMIP5 models are of the same sign.

Figure 3 Difference of ensemble mean (a) SST between the first (1-20 years) and the last (121-140 years) 20 years in the quadrupling experiment from 21 CMIP5 models (the last 20 years minus the first 20 years). (b) is the same as in Fig. 3a except the SST anomaly obtained by subtracting tropical (30°N – 30°S) mean SST. Contours show ensemble mean SST and SST anomaly field for the first 20 years with intervals of 2 K and 1.5 K, respectively. Dotted denotes the region where the responses of 14 or more out of the 21 CMIP5 models are of the same sign.

Supplementary Figure 7 (a) The ensemble mean composited map of 200hPa velocity potential and divergent wind in the years when El Niño occurs during boreal winter (December-January-February) in the first 20-years from 21 CMIP5 model simulation

(quadrupling experiment of CO₂ concentration). (b) is the same as in (a) except SST anomaly. Contours in (b) denotes the ensemble mean SST during boreal winter in the first 20-years from 21 CMIP5 model simulation. Dotted in (a), (b) denotes the region where the responses of 14 or more out of the 21 CMIP5 models are of the same sign.

Supplementary Figure 8 Latitudinal distribution of ensemble mean of (a) annual precipitation and (b) 500 hPa vertical motion averaged in 0-360°E in the control run (i.e., 1-20 years) (red line). Black lines in (a), (b) indicate the difference of ensemble mean precipitation and 500hPa vertical motion (121-140 years minus 1-20 years), respectively. Note that green circles denote the region where the responses of 14 or more out of the 21 CMIP5 models are of the same sign.

2. The results presented in the manuscript are presented solely from the perspective of the CMIP5 ensemble mean. There is, however, a known large spread in the climate system response detected in the various models. The authors do not show any statistical analysis related to their results to back up the significance of the claims. Such investigation would be desirable at least (!) as supplementary material.

:Thank you for the reviewer’s careful comment. Responding to the reviewer’s comment, we examined how well do the ensemble members agree in the fields presented in the current study (please see Figs. 1-4 in the revised manuscript and additional supplementary figures). Responding to the reviewer’s comments, we add the following sentences in the revised

manuscript:

“Note that contour lines in Fig. 2a-c indicate the ensemble mean precipitation in the control run, which is identical to Fig. 1c and the precipitation response to QCO₂ is highly consistent among ensemble members (Fig 2a), which are not due to one or two models driving the maximum in precipitation change. Furthermore, it is indicated that QCO₂ results are more consistent among climate models, compared to other two experiments. It is also noteworthy that the precipitation differences between the ensemble mean of 21 CMIP5 pre-industrial runs and the response to QCO₂ (121-140 years) (Supplementary Fig. 3) are similar to that from the first to last 20- years in the quadrupling experiment of CO₂ (Fig. 2a). ”

Following the reviewer’s comments, the figure captions are revised as follows:

Figure 1. First 20-year ensemble mean distribution of (a) velocity potential function (colors) and divergent wind (arrow) at 200 hPa level, (b) divergent water vapor transport [Q_D], and (c) rain rate in the control run. Note that the zonal mean water vapor transport is removed in Fig. 1b. Dotted denotes the region where the responses of 14 or more out of the 21 CMIP5 models are of the same sign.

Figure 2. Difference distribution of ensemble mean annual precipitation for the (a) quadrupling, (b) tripling and (c) doubling of CO₂ concentration. Contours represent ensemble mean field of precipitation for the first 20 years with intervals of 2 mm day⁻¹ in each experiment. Dotted denotes the region where the responses of 14 or more out of the 21 CMIP5 models are of the same sign.

Figure 3 Difference of ensemble mean (a) SST between the first (1-20 years) and the last (121-140 years) 20 years in the quadrupling experiment from 21 CMIP5 models (the last 20 years minus the first 20 years). (b) is the same as in Fig. 3a except the SST anomaly obtained by subtracting tropical (30°N – 30°S) mean SST. Contours show ensemble mean SST and SST anomaly field for the first 20 years with intervals of 2 K and 1.5 K, respectively. Dotted denotes the region where the responses of 14 or more out of the 21 CMIP5 models are of the same sign.

Figure 4. Difference distribution (121-140 years minus 1-20 years) of ensemble mean of (a) velocity potential function (colors) and divergent wind (arrow) at 200 hPa level and (b) effective wind [V_E]. Contours in (a) indicate ensemble mean field of velocity potential for the first 20 years with intervals of $20 \times 10^5 \text{ m}^2 \text{ s}^{-1}$. Dotted denotes the region where the responses of 14 or more out of the 21 CMIP5 models are of the same sign.

3. Please elaborate on the choice of the first 20 years of the 1pctCO2 runs as a control state. As in this period the climate system is already responding to the additional CO2, why not use a mean over the piControl runs as a control state?

:Thank you for the reviewer's constructive comments. Responding to the reviewer's comments, we also examined the 21 CMIP5 pre-industrial run to confirm the results obtained from a quadrupling experiment of CO2 concentration. We add the following sentences in the revised manuscript.

“It is found that the spatial structures of velocity potential along with 200hPa divergent wind, Q , and the precipitation rate obtained from the ensemble mean of 21 CMIP5 pre-industrial runs are not much different from those in the control run (Supplementary Fig. 1). This similarity suggests that climate system is not meaningfully perturbed by the additional CO₂ in the control run (i.e, the first 20-year period), therefore, the difference between the two periods (1-20 year vs. 121-140 year) is considered to be the atmospheric responses to QCO₂.”

“It is also noteworthy that the precipitation differences between the ensemble mean of 21 CMIP5 pre-industrial runs and the response to QCO₂ (121-140 years) (Supplementary Fig. 3) are similar to that from the first to last 20- years in the quadrupling experiment of CO₂ (Fig. 2a).”

Supplementary Figure 1 Same as in Fig. 1, but derived from the ensemble mean of 21 CMIP5 pre-industrial runs in the last 250 years simulation period.

Supplementary Fig. 3 Difference of ensemble mean annual precipitation for the last 20 years (121-140 years) in the quadrupling experiment of CO₂ concentration and the last 250 years simulation period in the CMIP5 pre-industrial run. Contours represent ensemble mean annual precipitation for the last 250 years simulation period in the CMIP5 pre-industrial run with intervals of 2 mm day⁻¹. Dots denote the region where the responses of 14 or more out of the 21 CMIP5 climate models are of the same sign.

Minor comments:

1. Lines 54, 213: "... CO₂ increases in a warmer climate": this study does not treat the carbon cycle, therefore such a phrasing is misleading, as the 1pctCO₂ experiment treats the warmer climate as a result of CO₂ increase. Please reformulate.

:Responding to the reviewer's comment, the above sentence is revised as follows:

"Therefore, it is crucial to understand the details of the physical processes playing key roles in determining the spatial pattern of tropical Pacific precipitation in the warmer climate, due to the increase of greenhouse gas concentration."

2. Line 71: "... CMIP5 CO₂ experiment results...": there are several idealized climate experiments, please state which one is referred here.

: Corrected.

3. Lines 76-77: "... more increases in the western tropical Pacific and less increases in the eastern tropical Pacific": please reformulate to explain the increase of what parameter is referred here. Also, please explain what do you mean by "more increases": more regions of increase or a stronger rate of increase?

:Following the reviewer's comments, we revised as follows:

"In this study, using a single metric of water vapor transport in the tropics, we demonstrate that the CO₂ increase tends to break up the trans-Pacific Walker circulation into more regionally confined circulations in smaller scales, resulting in greater increases in precipitation over the western tropical Pacific as compared to the eastern tropical Pacific."

4. Line 80: "CO₂ emissions": 1pctCO₂ is not an emission scenario, suggested reformulation to "CO₂ concentration".

: Corrected.

5. Lines 81-82: "the impact of major anthropogenic GHGs": in the previous sentence the authors state that only CO₂ varies, therefore such a formulation is contradictory.

:The above sentence is revised as follows: "To underpin the impact of CO₂ concentration increase on the tropical circulation"

6. Lines 144-149: The argumentation is unclear: the authors have not investigated the individual impact of CO₂ increase on the circulation/precipitation, as the climate system

adjusts to the increased concentrations. Hopefully, a rephrasing would clarify this issue. Also, a strengthened Walker circulation would not necessarily be confined in the same zonal/meridional boundaries and would have a more zonally expanded pattern of increased precipitation (with more increase over the Maritime Continent). Therefore, the phrase in lines 148-149 would benefit from more clarification.

:Thank you for the reviewer's useful comments. Following the reviewer's, we further analyzed outputs from doubling and triple experiments of CO₂ concentration from 21 CMIP5 model simulations to confirm the results obtained from a quadrupling experiment of CO₂ concentration. We add the following sentences with a new figure in the revised manuscript.

“We also analyze outputs from doubling and tripling CO₂ concentration experiments from 21 CMIP5 model simulations (Methods section) and 21 CMIP5 pre-industrial simulations (Supplementary Table 1) to ensure that the results based on the quadrupling experiment of CO₂ concentration are representative.”

“It should be noted that using CMIP5 model simulations, the geographical distribution of the CMIP5 model ensemble mean response in the total precipitation for doubling and tripling of CO₂ concentrations¹¹ is quite similar to that due to QCO₂ (Figs. 2a-c). This result indicates that the spatial structures of tropical rainfall change are not much influenced by the magnitude of atmospheric CO₂ concentration change, implying that a similar mechanism may work for the doubling, tripling and quadrupling of CO₂ concentrations in a warmer world. However, it is noted that the degree of inter-model agreement is higher for QCO₂ experiment”

In addition, we revised the earlier version of the manuscript as follows:

“The change of the precipitation rate over the tropical Pacific to QCO₂ (Fig. 2a), i.e., greater increases in the western equatorial Pacific as compared to the central-to-eastern tropical Pacific, may be caused by an increased intensity of Walker circulation with a more zonally expanded pattern. This seems contradictory to the expected weakening of the Walker circulation under the global warming conditions found in most coupled climate models^{36-37,39-42}.”

7. Lines 167-173: Please add reference to the Fig. 3, where needed.

: Added.

8. *Line 181: After "... across the Pacific from 150 degE" it would be clearer is "eastward" is added.*

: Added.

9. *Lines 189-190: "... the area showing less precipitation change with negative or positive sign": What does the sign refer to?*

: The above sentence is revised as follows:

“The strongest decrease in the velocity potential roughly corresponds to the area showing the decreased precipitation over the western part of the Maritime continent and the eastern Indian Ocean (see Fig. 2a).”

10. *Lines 353-373: Weren't these metrics introduced in previous studies?*

: We add proper references in the revised manuscript.

11. *The manuscript would highly benefit from a proof-read made by a native speaker.*

: Thank you for the reviewer's careful comments. We appreciate it. Responding to the reviewer's comments, we carefully read the earlier version of the manuscript and improved their writing

12. *The format of the references should be consistent, e.g. see referenced 17, 18, 20, as well as removal of typos, e.g. reference 35.*

: Corrected.

13. *Suppl. Fig. 1: Please remove "(control)" from the titles of the subplots.*

: Corrected.

14. *Suppl. Fig. 2: Either refer to the Lau et al. 2013 paper without including the figure or adapt the figure in the style of the previous ones from the manuscript.*

: We remove this figure in the revised manuscript. .

References:

Ma, S. and Zhou, T. (2016). Robust strengthening and westward shift of the tropical Pacific Walker circulation during 1979–2012: A comparison of 7 sets of reanalysis data and 26 CMIP5 models. Journal of Climate, 29(9):3097–3118.

McGregor, S., Timmermann, A., Stuecker, M. F., England, M. H., Merrifield, M., Jin, F.-F., and Chikamoto, Y. (2014). Recent Walker circulation strengthening and Pacific cooling amplified by Atlantic warming. Nature Climate Change, 4(10):888–892.

Bayr, T., Dommenges, D., Martin, T., and Power, S. B. (2014). The eastward shift of the Walker circulation in response to global warming and its relationship to ENSO variability. Climate Dynamics, 43(9-10):2747–2763.

Byrne, M. P. and Schneider, T. (2016). Narrowing of the ITCZ in a warming climate: Physical mechanisms. Geophysical Research Letters, 43(21): 11350–11357.

Optional:

He, J. and Soden, B. J. (2015). Anthropogenic weakening of the tropical circulation: The relative roles of direct CO₂ forcing and sea surface temperature change. Journal of Climate, 28(22):8728–8742.

Su, H., Jiang, J. H., Zhai, C., Shen, T. J., Neelin, J. D., Stephens, G. L., and Yung, Y. L. (2014). Weakening and strengthening structures in the Hadley circulation change under global warming and implications for cloud response and climate sensitivity. Journal of Geophysical Research: Atmospheres, 119(10):5787–5805.

Reviewer #1 (Remarks to the Author):

The authors have addressed my concerns. I recommend accept.

Reviewer #2 (Remarks to the Author):

I appreciate the effort made by the authors to address the comments of both reviewers. The manuscript has now a much clearer line of thought and the novelty of the study is well presented in the Introduction, along with the relevance to previous studies. I am grateful for the update of the figures, which now share one style and make the message easier to comprehend.

Following these revisions, I recommend the manuscript for publishing. I still have some minor comments, but I do believe they can be addressed very quickly during a last proof-read of the manuscript:

Line 23: There seems to be a word missing. I suggest: "Understanding the spatial pattern of precipitation response to CO2 emission increases..."

Line 28: Please change "triple" to "tripling".

Line 90: I suggest "1% CO2 experiment outputs" instead of "1% CO2 experiments".

Lines 109-110: The authors write "establish a path mostly from the sinking region to ascending region over the tropics". I would suggest to localize these regions, e.g. "establish a path mostly from the sinking region in the subtropics to the ascending region over the Equator".

Line 129: As the perturbation, though not meaningful in the first 20 years of 1pctCO2, still exists, I propose to write: "the difference between the two periods ... is considered representative of the atmospheric responses to QCO2."

Line 335: Please change the order to "Doubling and tripling experiments ..." for consistency with the text below.

REVIEWERS' COMMENTS:

Reviewer #1 (Remarks to the Author):

The authors have addressed my concerns. I recommend accept.

Reviewer #2 (Remarks to the Author):

I appreciate the effort made by the authors to address the comments of both reviewers. The manuscript has now a much clearer line of thought and the novelty of the study is well presented in the Introduction, along with the relevance to previous studies. I am grateful for the update of the figures, which now share one style and make the message easier to comprehend.

Following these revisions, I recommend the manuscript for publishing. I still have some minor comments, but I do believe they can be addressed very quickly during a last proof-read of the manuscript:

:Thank you for the reviewer's careful comments. We appreciate it. Following the reviewer's comment, we revised the manuscript accordingly.

Line 23: There seems to be a word missing. I suggest: "Understanding the spatial pattern of precipitation response to CO2 emission increases..."

:Corrected.

Line 28: Please change "triple" to "tripling".

:Changed.

Line 90: I suggest "1% CO2 experiment outputs" instead of "1% CO2 experiments".

:Corrected

Lines 109-110: The authors write "establish a path mostly from the sinking region to ascending region over the tropics". I would suggest to localize these regions, e.g. "establish a path mostly from the sinking region in the subtropics to the ascending region over the Equator".

:Corrected.

Line 129: As the perturbation, though not meaningful in the first 20 years of 1pctCO2, still exists, I propose to write: "the difference between the two periods ... is considered representative of the atmospheric responses to QCO2."

:Corrected.

Line 335: Please change the order to “Doubling and tripling experiments ...” for consistency with the text below.

:Corrected.